# OpenMedLM: Prompt engineering can out-perform fine-tuning in medical question-answering with open-source large language models

Jenish Maharjan*[†], MS, Anurag Garikipati*[†], MS, Navan Preet Singh[†], MS, Leo Cyrus[†], PhD, Mayank Sharma[†], MS, Madalina Ciobanu[†], PhD, Gina Barnes[†], MPH, Qingqing Mao[†], PhD, Ritankar Das[†], MSc

*Joint first authors [†]All authors were employees of Montera, Inc dba Forta at the time of completion of the study

**Introduction:** LLMs have become increasingly more capable at a growing range of language tasks, and they inherently constitute a powerful tool that can expand equitable access to medical knowledge.[1] This has promoted interest in the integration of LLM-based AI tools within various medical tasks, such as diagnostics aids, and tools developed for patient use to help with understanding care plans.[2-5] However, prior to deploying LLMs into real-world medical settings, models intended for healthcare implementation have to be proven to be accurate, unbiased, and safe for use with patients. As a first step to successfully developing healthcare LLMs, models have to be evaluated on medical benchmarks developed for performance evaluation of medically-specialized LLMs.[6-9] Some common benchmarks include MedQA, MedMCQA, PubMedQA, and the medical-subset of MMLU (nine medically and clinically relevant subsets).[6-9]

Optimizing performance of healthcare LLMs on medical benchmarks can entail a variety of approaches, with the majority involving some degree of fine-tuning, which may be accompanied by expensive computational costs that are out of reach for most and may require task-specific data that is not easily accessible. While proprietary models have achieved strong performance, the ability to perform further investigation into the model's performance is limited. Adapting the techniques used in proprietary model optimization to open source (OS) models can provide the broader research community with a deeper understanding of the approaches, while retaining the benefits of OS models including transparency and compliance, which are particularly important in the healthcare space.

While fine-tuning has produced strong results on evaluating LLMs on medical benchmarks, recent studies have examined employing robust prompt engineering to optimize performance of foundation models to similar or better levels when tested on medical benchmarks.[10,11] For example, Microsoft developed Medprompt, a robust prompting technique, for use with the generalist GPT-4 model and achieved SOTA results the most common medical Q&A benchmarks.[10]

However, there is no study to date showing that robust prompt engineering can be applied to generalist OS foundation models to significantly optimize performance in the absence of specialized fine-tuning. In this study, we present the OpenMedLM prompting platform, which applies robust prompt engineering techniques to the OS Yi 34B foundation model to achieve SOTA results on the four medical benchmarks we evaluated: MedQA, MedMCQA, PubMedQA, and the medical-subset of MMLU. By employing a range of prompting techniques including few-shot prompting,[12] chain-of-thought (CoT) prompting,[13] and self-consistency,[14] our OpenMedLM prompting platform when used with the OS Yi 34B[15] foundation model outperformed Meditron,[1] the previous SOTA for OS LLMs, on the MedQA (Figure 1), MedMCQA, and the medical-subset of the MMLU benchmarks.

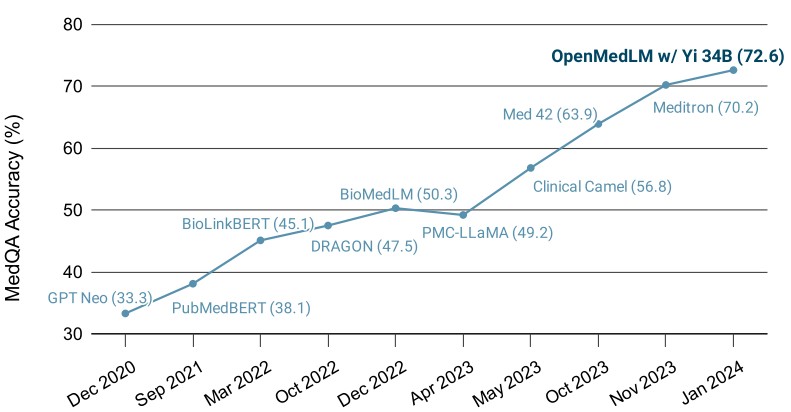

Figure 1: OpenMedLM performance on MedQA benchmark. OpenMedLM achieves 72.6% accuracy on the MedQA dataset with the Yi 34B model, surpassing all other OS models.

**Methods:** Due to our focus on multiple choice Q&A benchmarks, we use accuracy as the evaluation metric to compare performance. During the inference for evaluation, we allowed a maximum of 5 tries for the model to generate a valid output, defined as an output that could be interpreted as an answer to a multiple choice question. If the model did not generate a valid output after 5 tries, the answer was counted as incorrect and would reflect as such in the final computation of accuracy for the benchmark. As the maximum number of tokens for the model also affects the GPU memory usage and the time for evaluation, we selected a value for the maximum number of tokens empirically to mimic the average length of the tokens in the CoT explanations of the questions in the examples used in the prompt. The instructions in the prompts were also selected empirically to optimize performance. A series of prompt instructions were tested to the determine instructions that maximized performance.

As a variation of the CoT examples, we implemented a k-nearest neighbors (kNN) method, based on the approach described by Nori et al.,[10] to select the most similar questions from the respective training sets for every test question. The kNN approach finds the 5 nearest neighbors to the test question in the dataset's training split, creating unique prompts for each test question to best encompass the context of that particular test question. Additionally, an ensemble voting scheme was

implemented, which runs each prompt through the model 5 times and selects the answer most commonly output during the 5 runs. For each of the 5 inference runs, the multiple choice options were randomly shuffled to create variation in the order in which the correct option was presented. An ablation experiment was performed to evaluate the model's performance on each prompting technique individually prior to adding on the subsequent prompting techniques. Results were computed for each prompting strategy, as well as for the complete OpenMedLM prompting platform in order to determine the accuracy of the prompt engineering approach as applied to the OS Yi 34B foundation model.

**Results:** Through the combination of all sub-components of the OpenMedLM prompting platform, we showcase SOTA performance on 3 of the 4 medical Q&A benchmarks evaluated in this study. We performed an ablation experiment which highlights the contribution of each sub-component of the OpenMedLM prompting platform to the overall results of the model. Figure 2 showcases the effect of each sub-component of OpenMedLM for the MedQA benchmark, by comparison with the compounding effect of each prompting technique employed by Meditron to achieve their model's top accuracy. The combination of all components of the OpenMedLM prompting platform results in a prompt engineering approach that enables the Yi 34B foundation model to outperform the specialized Meditron model.

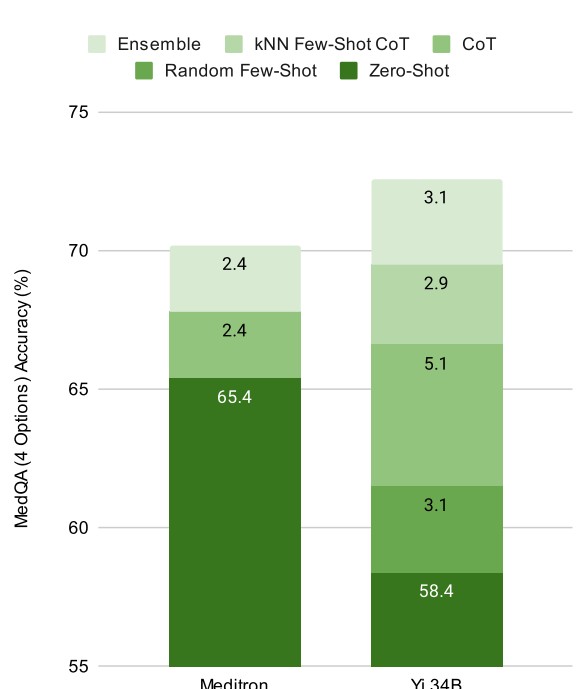

Figure 2: Ablation study showing the contribution of different components of the OpenMedLM prompting platform vs. the Meditron fine-tuned model.

*Zero-Shot*: The baseline performance of the Yi 34B foundation model absent any prompt engineering techniques achieves an accuracy of 58.4% on the MedQA dataset.

*Random Few-Shot*: Following the implementation of random few-shot prompting, the performance of the Yi 34B model increased by 3.1% to deliver an accuracy of 61.5%.

*Random Few-Shot with CoT*: CoT explanations were utilized for each of the few-shot examples and added as the input prompt to the Yi 34B foundation model. This prompting strategy resulted in the largest surge in performance, contributing to a 5.1% increase leading to an accuracy of 66.6% for the Yi 34B foundation model.

*kNN Few-Shot with CoT*: To further test the importance of the specific examples utilized in few-shot prompting, a kNN algorithm was used to identify the 5 most similar training questions to each test question and generate a CoT explanation for each of those examples to construct the prompt. The kNN few-shot with CoT approach led to a 2.9% improvement in accuracy on MedQA to deliver an accuracy of 69.5% for the Yi 34B foundation model.

*Ensemble/Self-Consistency*: The final prompting strategy that we employed utilized a self-consistency approach to develop a majority voting scheme for answer selection. The voting scheme provided an additional 3.1% improvement in performance which led to the achievement of an overall total accuracy of 72.6% for the Yi 34B foundation model.

**Discussion:** We present OpenMedLM, a prompting platform which enables SOTA results for OS foundation LLMs on common medical benchmarks. OpenMedLM facilitates SOTA level performance solely by utilizing robust prompt engineering on generalist OS foundation LLMs. Through a series of additive and synergistic prompting techniques, OpenMedLM enables performance superior to previous SOTA on 3 of the 4 evaluated medical benchmarks for the Yi 34B foundation model we employed, which showcases the potential of generalist OS foundation models to perform highly specialized tasks without the costly challenges of requiring a highly-specialized training or fine-tuning dataset that are necessary to develop specialized models.

Achieving best performing results on specialized tasks by employing generalist OS foundation models highlights the need to continue research into better understanding the full extent of the capabilities of these OS foundation models. Research into LLMs has shown that large models have potential emergent abilities which may not have been thought of when the model was trained.[16] The ability to achieve high accuracy on medical Q&A with foundation models highlights the emergence of properties for healthcare specific tasks which were previously assumed to require fine-tuning. The ability of OpenMedLM to bring out emergent properties in OS models highlights the need for further research into the full capabilities of generalist OS foundation LLMs on medical tasks, which we intend to further research going forward.

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

**Appendix:**

Results of Ablation Study:

**Table A1: Performance of top-performing OS models on multiple-choice medical benchmarks.** OpenMedLM delivers state-of-the-art results on 3 of the 4 primary benchmarks with the Yi 34B foundation model. Results for the specialized Meditron model were sourced from Chen et. al.[1] (ZS = zero-shot; FS = few-shot; CoT = chain-of-thought; kNN = k-nearest neighbors).

| Model | Benchmark Accuracy | | | |
|---|---|---|---|---|
| | MedQA (4 Options) | MedMCQA | PubMedQA | MMLU - Medical |
| *ZS* | | | | |
| Yi 34B | 58.4 | 55.9 | 53.4 | 72.6 |
| Meditron 70B | 65.4 | 65.1 | 80.0 | 73.6 |
| *Random FS* | | | | |
| Yi 34B | 61.5 | 58.0 | 57.0 | 77.3 |
| *CoT* | | | | |
| Yi 34B (FS) | 66.6 | 59.2 | 68.2 | 75.2 |
| Meditron 70B (ZS) | 67.8 | 63.2 | 81.0 | 74.9 |
| *kNN FS CoT* | | | | |
| Yi 34B | 69.5 | 65.7 | 72.4 | 78.7 |
| *Ensemble/Self-Consistency* | | | | |
| Yi 34B (kNN FS w/ Shuffling) | **72.6** | **68.3** | 77.3 | **81.6** |
| Meditron 70B (Random FS) | 70.2 | 66.0 | **81.6** | 77.6 |

Table 1 highlights the performance of the Yi 34B foundation model with the complete OpenMedLM prompting platform (i.e., employing all prompting strategy components of OpenMedLM) compared against the performance of the Meditron 70B specialized model, which achieved the previous SOTA for OS LLMs. Table 1 further shows the breakdown of results of each prompting strategy in an ablation experiment. While the addition of each successive prompt results in an additive effect on the accuracy of the Yi 34B model for the MedQA, MedMCQA, and PubMedQA benchmarks, the combination of prompts results in a synergistic effect on the accuracy of the model on the medical-subset of the MMLU benchmark.

OpenMedLM Implementation:

When experiments were performed to evaluate performance, the prompting techniques were run sequentially starting with zero-shot prompting, where each subsequent prompting approach was added on to previous techniques as an ablation study as described below:

i.     Zero-shot prompting was run solely with an instruction followed by the question and multiple choice options.

ii.    Few-shot prompting was run with an instruction followed by random selection examples consisting of questions and corresponding answers from the dataset. The question to be answered with corresponding options followed the randomly selected examples.

iii.    The CoT prompting with the random few-shot examples was presented with an instruction, followed by an example question with possible answers, and then an explanation (CoT reasoning) along with the correct answer, subsequently followed by the question to be answered and its possible answers. For the CoT prompts, we generated the CoT explanations using GPT-4 via the OpenAI API.[17] Each of the questions selected to be used as examples in the CoT prompts was prompted to GPT-4 for generating a CoT explanation along with the correct answer. Fig. A1 shows the template for generating the CoT explanation which was provided to GPT-4.

iv.    The CoT prompting with the kNN-based selection of few-shot examples followed a similar template for generating explanations as the CoT with random few-shot examples. However, as explanations needed to be generated for each of the 5 examples for each question, a caveat was added for incorrect answers during generation of the CoT explanation. If the answer was incorrect, the question was resubmitted up to 3 times for the correct answer to be generated. In cases where GPT-4 was not able to generate a correct answer from 3 trials, the question was replaced with the next most similar question from the training set based on kNN output.

```
You are a medical expert. Answer the following multiple choice
question from the medical domain based on following instructions.
1. Output a brief explanation summarizing and providing context to
the question under the heading 'Explanation' in about 5 sentences.
2. Select the correct option and provide the correct option under
the heading 'Answer'.
3. Always select one of the four options provided as the answer.
4. If the options are ambiguous or the question does not have enough
context, select the one that best answers the question.

### Question: {question}
### Options: {options}
```

**Fig. A1: The prompt fed to GPT-4 to generate the chain-of-thought (CoT) explanations for each few-shot example for both the random few-shot examples and the kNN-based few-shot examples.** For kNN CoT, the prompt was run up to 3 times per example to achieve a correct answer before the next most similar example was fed into GPT-4.

