# OpenReview forum: "OpenMedLM: Prompt engineering can out-perform fine-tuning in medical question-answering with open-source large language models"
_AAAI.org/2024/Spring_Symposium_Series/Clinical_FMs — AAAI 2024 SSS on Clinical FMs_

### Official Review · Reviewer_bmaT · 2024-02-17
**The paper effectively demonstrates the superiority of various prompting techniques by showcasing their practical applicability to open-source models. However, improvements are needed in paper formatting, figure clarity, wider evaluation, and code release to enhance the paper's overall impact and reproducibility.**

**Rating:** 6
**Confidence:** 5

**Review:**

**Strengths:**
- The authors effectively demonstrate how various prompting techniques can lead to superior performance compared to models finetuned/pretrained on domain-specific data. This experimentation showcases the practical utility of these techniques for open-source models, benefiting the community.
- The paper provides clear and comprehensive background details, making it easy to follow.

**Weaknesses:**
- It is essential for the authors to include an abstract and adhere to the formatting guidelines specified in the AAAI-24 Author Kit.
- The authors have failed to cite papers referenced in Figure 1.
- Figure 2 suffers from clarity issues, possibly due to colour choices. Reproducing results from open-sourced models like Meditron would ensure consistency in the computing environment. The same applies to models such as Med42, Clinical Camel, and PMC-LLaMA.
- The evaluation of the authors’ claims is limited. Experimentation across different model architectures and sizes would provide stronger support for their assertions.
- The lack of code release for their OpenMedLM prompting platform needs to be addressed.

---

### Official Review · Reviewer_duC5 · 2024-02-21
**Weak Accept**

**Rating:** 6
**Confidence:** 3

**Review:**

The paper introduces innovative prompting techniques and demonstrates their effectiveness in improving the performance of the general language model Yi-34B over a clinically fine-tuned LLM, Meditron 70B, across three out of four medical Q&A benchmarks: MedQA (4 Options), MedMCQA, PubMedQA, and MMLU - Medical.

**Pros**

1) **Clarity of paper**: The paper is well-written and easy to follow. Figure 1. showing LLM performance with time is good.

2) **Good improvements and ablations**: The results with different prompting techniques are well demonstrated in Figure 2 and Table A1.


**Cons**

1) **Unavailability of training dataset for KNN FS CoT (minor)**: Since this requires access to train set samples, it is a costly process, and training data won't always be available for many models. I do understand the point of open source LLMs which the paper advocates. But consider medical domain datasets that are sensitive to release. Hence the authors are not able to show results of Meditron 70B on kNN FS CoT. This is to be discussed in limitations.

2) **Experiment baselines**: The exclusive focus on Meditron 70B as a baseline leaves the comparison somewhat narrow. Including at least one additional medical LLM baseline would offer a broader perspective on the presented techniques' relative performance.

3) **Missing Proper References**: The paper lacks references for the various prompting methods (CoT, KNN prompting, etc.) utilized in the study. These missing references raises following two questions:

4) **Novelty**: : It is unclear whether the use of KNN with training samples is a novel contribution of this paper or if it has been previously used.

5) **How are similar questions selected for kNN FS CoT?**
I didn't find how similar questions to test examples are chosen via KNN in the paper. It is done in which space (embedding space of model/input space). Please explain it properly.

6) **Limitations of kNN FS CoT and other prompting strategies:**  The potential for prompting strategies, such as kNN FS CoT, to overemphasize reasoning patterns from similar training samples needs further discussion. Highlighting this and other limitations of the proposed methods would provide a more balanced and comprehensive view of their applicability.

7) The clarity of Figure 2. can be improved with the usage of other shades of colors than green for more clarity.

The paper presents evidence that innovative prompting strategies can enhance the performance of open LLMs compared to fine-tuned counterparts in the medical domain. However, addressing the aforementioned concerns would strengthen the paper.

---

### Official Review · Reviewer_irrN · 2024-02-21
**Enhancing Medical Question-Answering with Prompt Engineering**

**Rating:** 6
**Confidence:** 3

**Review:**

OpenMedLM proposes a platform employing prompt engineering techniques to optimize the performance of open-source large language models (LLMs) on medical question-answering benchmarks, achieving state-of-the-art (SOTA) results without fine-tuning. Using the Yi 34B model, OpenMedLM surpasses previous SOTA results on MedQA, MedMCQA, PubMedQA, and MMLU medical subsets through few-shot prompting, chain-of-thought (CoT) prompting, and self-consistency strategies. The study emphasizes the potential of prompt engineering in enhancing the capabilities of generalist LLMs for specialized tasks like medical question answering.

**Strengths**
- Innovative Approach: The study introduces a novel use of prompt engineering as a viable alternative to fine-tuning, offering a cost-effective method for enhancing LLM performance in specialized domains.
Comprehensive Evaluation: It evaluates the model across multiple benchmarks, thoroughly assessing its capabilities in medical question-answering tasks.
- Clear Methodology: The methodology, including using few-shot prompting, CoT prompting, and self-consistency, is well-articulated, offering clarity on the process and potential for replication.

**Weaknesses**
- Generalization Concerns: The study's focus on a single LLM (Yi 34B) raises questions about the generalizability of the findings to other models.
- Lack of Comparative Analysis: While it compares OpenMedLM's performance with that of the Meditron model, a broader comparison with more models, especially those using different fine-tuning approaches, could have provided a more comprehensive view of its relative performance.

---

### Official Review · Reviewer_LwgB · 2024-02-26
**Prompt engineering for open-source large language models**

**Rating:** 7
**Confidence:** 4

**Review:**

This paper first demonstrated that prompt engineering can outperform fine-tuning in medical QA for open-source models. I really like this work and definitely think it's important as open-source models are more accessible for clinicians and have less privacy issues. Overall I think this is a good paper presenting important conclusions. My only concern is that this paper heavily relies on public benchmark and lacks the evaluation from the clinician.